# Similarities in Virulence and Extended Spectrum Beta-Lactamase Gene Profiles among Cefotaxime-Resistant *Escherichia coli* Wastewater and Clinical Isolates

**DOI:** 10.3390/antibiotics11020260

**Published:** 2022-02-17

**Authors:** Elizabeth Liedhegner, Brandon Bojar, Rachelle E. Beattie, Caitlin Cahak, Krassimira R. Hristova, Troy Skwor

**Affiliations:** 1Department of Biomedical Sciences, College of Health Sciences, University of Wisconsin-Milwaukee, Milwaukee, WI 53211, USA; liedhegn@uwm.edu (E.L.); bbojar@uwm.edu (B.B.); 2Department of Biological Sciences, Marquette University, Milwaukee, WI 53233, USA; rachelle.beattie20@gmail.com (R.E.B.); krassimira.hristova@marquette.edu (K.R.H.); 3Wisconsin Diagnostic Laboratories, Milwaukee, WI 53226, USA; ccahak@wisconsindiagnostic.com

**Keywords:** wastewater, ESBL, KPC, CTX-M, virulence, uropathogenic

## Abstract

The World Health Organization has identified antibiotic resistance as one of the largest threats to human health and food security. In this study, we compared antibiotic resistance patterns between ESBL-producing *Escherichia coli* from human clinical diseases and cefotaxime-resistant environmental strains, as well as their potential to be pathogenic. Antibiotic susceptibility was tested amongst clinical isolates (*n* = 11), hospital wastewater (*n* = 22), and urban wastewater (*n* = 36, both influent and treated effluents). Multi-drug resistance predominated (>70%) among hospitalwastewater and urban wastewater influent isolates. Interestingly, isolates from clinical and urban treated effluents showed similar multi-drug resistance rates (~50%). Most hospital wastewater isolates were Phylogroup A, while clinical isolates were predominately B2, with a more diverse phylogroup population in urban wastewater. ESBL characterization of cefotaxime-resistant populations identified *bla*_CTX-M-1_ subgroup as the most common, whereby *bla*_KPC_ was more associated with ceftazidime and ertapenem resistance. Whole-genome sequencing of a carbapenemase-producing hospital wastewater *E. coli* strain revealed plasmid-mediated *bla*_KPC-2._ Among cefotaxime-resistant populations, over 60% of clinical and 30% of treated effluent *E. coli* encoded three or more virulence genes exhibiting a pathogenic potential. Together, the similarity among treated effluent *E. coli* populations and clinical strains suggest effluents could serve as a reservoir for future multi-drug resistant *E. coli* clinical infections.

## 1. Introduction

Antimicrobial resistance (AMR) is a serious global health concern. In the United States alone, over 2 million AMR infections are reported annually, resulting in approximately 23,000 deaths [1]. Although AMR is frequently associated with the overuse and misuse of antibiotics in clinical settings, resistant bacteria and their genes are widespread within human, animal, and environmental settings, indicating the need for a One Health approach [2]. Wastewaters are ideal for the study of AMR as they serve as a bridge between human, animal, and environmental settings and have been frequently implicated in the acquisition and dissemination of AMR bacteria and genes [3,4]. Additionally, wastewaters are rich in selective pressures including antibiotics, pharmaceuticals, and biocides that can increase microbial horizontal gene transfer (HGT) and mutation rates, leading to an increase in AMR [5,6].

Wastewaters contain bacteria from a variety of sources including human waste, stormwater, and hospital sewage. Due to the widespread use of broad-spectrum antibiotics for disease treatment in hospital settings, hospital wastewater frequently contains elevated levels of antibiotic residues and antibiotic resistant bacteria, many of which are resistant or resilient to wastewater (WW) treatment processes [7,8] and enter urban waterways and recreational waters [9,10]. Bacterial populations that survive the WW treatment process, whether pathogenic or not, serve as a health risk in downstream receiving waters by increasing the dissemination of AMR bacteria and their genes.

Specific classes of hospital-administered antibiotics pose greater human health risk if disseminated into the environment. One such last-resort antibiotic class, third-generation cephalosporins, is frequently used as a front-line therapeutic against serious Gram-negative bacterial infections [1]. Increased activity beta-lactams comprise 10.6% of all prescribed antibiotics in the United States [11]. However, The United States Center for Disease Control and Prevention (CDC) has identified strains of the family Enterobacterales resistant to this class of antibiotics as a major public health threat [1]. Resistance to cephalosporins is mediated by a diverse group of antibiotic resistance genes (ARGs) called extended spectrum beta-lactamase (ESBLs) [12]. ESBL-producing *E. coli* populations are defined by resistance to penicillins, cephalosporins, and aztreonam that can be inhibited by beta-lactamase inhibitors like clavulanic acid [13]. These ESBLs are highly associated with mobile genetic elements capable of being passed to other bacteria via horizontal gene transfer (HGT) [14], which can be elevated by many stressors present in the WW [15]. A better understanding of the contribution of hospital WW to the presence and dissemination of ESBLs throughout the WW treatment system is necessary to determine human health risk in downstream receiving waters, as well as further assessing the potential of resistant populations to be pathogenic. 

*Escherichia coli* populations comprise both pathogenic and commensal strains and are strong indicator species of warm-blooded animal and human fecal waste [16]. AMR populations of *E. coli* present two public health threats: increased difficulty to treat clinically and ability to transfer ARGs both intra- and interspecies [17], thus being a vehicle of ARG dissemination. Identifying similarities in AMR phenotypes and genotypes, as well as virulence factors between isolates from interconnected ecosystems, can help identify areas in which gene transfer may be occurring that are in critical need of AMR mitigation and control measures. In areas that release treated wastewater into local receiving waters accessed by humans and animals, this evaluation is vital to reduce potential health risks. Previous studies by our labs have characterized antibiotic resistance phenotypes and genotypes of *E. coli* from manure, wastewater, sediment and surface waters, hospital wastewater, and clinical strains [10,18,19]. This study expands on our previous findings through the following aims: (i) to assess the presence of cefotaxime (CTX)-resistant *Escherichia coli* populations in hospital and urban WW; (ii) to compare resistance patterns among clinical ESBL-producing *E. coli* to urban and hospital WW strains within a defined geographical location; and (iii) to further assess the presence of virulence factors among cefotaxime resistant populations to provide a more complete public health risk analysis. 

## 2. Results and Discussion

### 2.1. Characterization of Antibiotic Resistance Profiles of Cefotaxime-Resistant E. coli

Among the leading antimicrobial resistance threats facing healthcare, ESBL-producing Enterobacterales pose a serious health threat with higher healthcare costs and longer hospital stays [20]. Furthermore, the worldwide carriage rate of ESBL-producing *E. coli* populations has significantly increased over eight-fold (2.6% to 21.1%) in less than 15 years [21]. Combating this issue involves utilizing a One Health approach, recognizing that AMR is interconnected between the clinic, agriculture, and the environment. In this study, we examined ESBL-producing *E. coli* isolates from a variety of clinical sources (7- uropathogenic *E. coli* (UPEC), 2- bacteremia, 1- wound infection, and 1- surgical tissue infections) from Milwaukee, Wisconsin, U.S.A. All selected isolates exhibited at least intermediate resistance to third-generation cephalosporins or carbapenems. Clinical isolates were analyzed for their antibiotic resistance profiles and were compared to cefotaxime-resistant *E. coli* isolates from WW effluents of the same hospital (*n* = 22), as well as influents (*n* = 19) and treated effluents (*n* = 17) from urban wastewater reclamation facility in the same city. The Milwaukee wastewater reclamation facility services 1.1 million people within a 681 km^2^ area collecting and treating on average 265 mL/day of sewage during dry days. The sewage comprises domestic urban surface runoff, industrial, and hospital wastewater. The final treated wastewater is discharged into Lake Michigan within the Milwaukee Bay area (43.02329162715021, −87.89408612678749). Similar to a global analysis of disinfected WW effluents [22], no CTX-resistant *E. coli* isolates were identified in post-chlorinated WW effluents. The 69 *E. coli* isolates were tested for resistance against twelve different antibiotics. Hospital WW displayed the same or highest prevalence of antibiotic resistance compared to all other *E. coli* sources except for chloramphenicol (CHL), to which all isolates were fully susceptible (Figure 1). Although antibiotic resistance was not as prevalent among WW influents and effluents, more than 25% of isolates were resistant to five antibiotics (sulfamethoxazole-trimethoprim (SXT), tetracycline (TET), ceftriaxone (CRO), ceftazidime (CAZ), and CTX) (Appendix A).

Multi-drug resistance (≥3 antibiotic classes), a major health threat, was evident among over 71% (49 or 69) of isolates with the highest prevalence among hospital WW and urban influent WW (Figure 1: 91% and 74% respectively). Most CTX-resistant hospital WW *E. coli* were also resistant to SXT, gentamicin (GEN), and ciprofloxacin (CIP) (Appendix A), consistent with another report [23]. Resistance to these three antibiotics (SXT, GEN, and CIP), albeit at lower levels, was persistent throughout influent (68%, 58%, and 39%, respectively) and effluent (47%, 24%, and 12%, respectively). Although antibiotic resistant profiles were quite diverse within and among the sample sources (Appendix A), resistance to SXT, GEN, CIP, TET, CRO, CAZ, and CTX were the most prevalent among hospital and urban influent WW. Similar to other reports [24], hospital WW isolates had significantly higher multiple antibiotic resistance (MAR) indices than all other sources (Figure 2A: *p* < 0.05). 

In addition to antibiotic susceptibility, we also categorized these *E. coli* by phylogroup [25], which aids in identifying a possible common bacterial host or origin. Most clinical isolates (64%) were attributed to phylogroup B2 (Appendix A), which correlated with the abundance of uropathogenic strains [26] amongst our clinical population. Interestingly, B2 was also present in WW influents (7%) and treated effluent (13%) isolates, though not observed in hospital WW (Appendix A). The majority of hospital WW was comprised of phylogroup A (77%), which typically associates as human commensal origin [27]. In agreement with other studies [19,28], WW influents and effluents showed a greater diversity in phylogroups (Appendix A) likely due to multiple sources combining into one, as well as WW persistent populations [27]. Amongst all the phylogroups, isolates typed to phylogroup A were significantly more resistant than those typed to B2 and D (Figure 2B, *p* < 0.05). This finding could be attributed to microbial source, as phylogroup A comprised most samples from hospital WW, similar to other reports [19,29,30].

### 2.2. Prevalence of β-Lactamases among CTX-Resistant Populations

Among CTX-resistant *E. coli* populations from the clinic and environmental sources, we assessed the presence of seven different β-lactamases: CTX-M subgroups (1, 2, and 9), TEM, SHV, OXA, and KPC. They have become a global pandemic with ESBL prevalence increasing among commensals and clinical isolates [21,31], causing significant therapeutic challenges [32]. Women with ESBL-producing *E. coli* urinary tract infections are significantly more likely to contain the same ESBL-producing *E. coli* clone in their feces [33]. From a One Health perspective, the strongest correlation of ESBL genes among commensal and pathogenic isolates is with environmental sources, specifically surface and WW [34], in contrast to agricultural sources. One of the most common ESBLs that continues to increase in prevalence is the CTX-M family [34,35]. CTX-M enzymes originally targeted cefotaxime; however, their predominant presence in mobile genetic elements increased the variability and hydrolytic activity to other cephalosporins, including ceftazidime [36]. Furthermore, intra- and interspecies HGT within WW poses an additional threat with the evolution of chimeric CTX-M genes [37]. In our study, CTX-M-1 subgroup was the most common among ESBLs (Table 1: 81% of total samples analyzed) where 95% of cefotaxime-resistant hospital WW populations encoded *bla*_CTX-M-1_ subgroup followed by 73% of urban WW influent and 81% of treated effluent isolates (Table 1). Resistance to CRO and CTX were significantly correlated with strains encoding *bla*_CTX-M-1_ (Figure 3, *p* = 0.0001). Among clinical isolates, the CTX-M-1 subgroup predominated as the second most common ESBL among clinical strains, where it was encoded by 64% of isolates (Table 1). The CTX-M-1 subgroup is the most common ESBL within hospital isolates from North America comprising between 34% and 66.2% [38,39]. Another study demonstrated 79% and 67% of ESBL-producing *E. coli* phylogroup B2 strains from urine and feces, respectively, encoded CTX-M-1 subgroup compared to 21% and 25% CTX-M-9 subgroup [33]. 

The CTX-M-9 subgroup was more prevalent among urban WW isolates compared to clinical and hospital WW (Table 1, *p*= 0.005). Although not as common as CTX-M-1 subgroup, the CTX-M-9 subgroup members were initially found on conjugative plasmids [40]. This subgroup, along with CTX-M-1, is commonly present within hospital sinks [41]. Others have found both CTX-M subgroups 1 and 9 at similar frequencies among WW influent *E. coli* populations [37]. Within the United States, CTX-M subgroups 1 and 2 have been located without subgroup 9, based on total WW DNA analysis [42]. Bevan et al. reviewed 220 articles about CTX-M subgroups and identified a predominate geographic distribution of CTX-M-1 and -9 subgroups among most of the world, though some countries in South America display a stronger CTX-M-2 subgroup population [43]. Our findings agree with the worldwide CTX-M-1 and -9 subgroup pattern and no CTX-resistant isolates encoding CTX-M-2 subgroup. With the advent of sequencing technologies providing more affordable sequencing options, it is important to distinguish between the health risks associated with ESBL findings from total DNA compared to bacterial isolates. 

Among the other β-lactamase genes analyzed in our study, *bla*_TEM_ and *bla*_OXA_ were carried by 73% of isolates (Table 1). Both ESBLs were found on a relatively high percentage in all sources, although clinical sources, like Branger et al. [44], had the highest prevalence of *bla*_TEM_ compared to hospital WW isolates, which had the lowest prevalence (Table 1: 100% vs. 59%, respectively). However, this pattern was reversed with *bla*_OXA_, where 86% of hospital WW isolates encoded it, though only 45% of clinical strains carried the gene (Table 1). Further analysis of *bla*_TEM_ and *bla*_OXA_ genotypes with resistance phenotypes did not identify any significant correlations (Table 2). Our study did not identify any *bla*_SHV_-encoding CTX-resistant strains from either hospital or urban WW_._ This is in agreement with other studies worldwide reporting a predominance in TEM, OXA, and CTX-M [45,46].

When assessing the risk of ARGs within environmental bacterial populations, it is important to assess if the ARGs encoding resistance against currently used antibiotics, especially last-resort antibiotics, are found on mobile genetic elements [47]. In our study, we have identified plasmid-mediated resistance to third-generation cephalosporins (Appendix A) from WW sources. Additionally, class 1 integron-integrase gene (*int1*) was found among 65% of all CTX-resistant populations with similar levels amongst clinical and WW populations (Table 1: 64.4% and 64.2% respectively). Our findings agree with others who identify *int1* commonly throughout WW treatment process [48], as well as harboring gene cassettes encoding ARGs [19], thus supporting its use as an indicator of arthropogenic pollution [49]. Widespread dissemination of *int1* among diverse bacterial populations within mobile genetic elements, such as plasmids and transposons, supports a strong association of their presence with the rapid evolution of antibiotic resistance [49].

With carbapenems traditionally used to treat ESBL-producing Enterobacteriaceae infections, the evolution of carbapenemases pose a direct threat to therapeutic options. Enterobacteriaceae-produced carbapenemases are primarily grouped into Ambler class A (*Klebsiella pneumoniae* carbapenemase (KPC) and GES), class B (metallo-β-lactamases (MBL) like NDM, VIM, and IMP), or class D (OXA-48-like enzymes) [50]. Of these, KPC-producers are the most endemic throughout the Americas, parts of Europe, and China, while spreading globally [51]. Amongst our cefotaxime-resistant *E. coli* populations, hospital WW was the only source that carried *bla*_KPC_ (Table 1: 41%, *p* < 0.001). The presence of *bla*_KPC_ was significantly correlated with carbapenem and ceftazidime resistance and the absence of *bla*_TEM_ (Table 2, *p* < 0.01). Other research has found that the highest prevalence of clinical *bla*_KPC_ appears within inpatient facilities, peaking in long-term acute care hospitals [51]. 

### 2.3. Whole-Genome Sequence Analysis of Carbapenemase-Producing E. coli Isolate

To further characterize the *bla*_KPC_-encoding populations, we sequenced one of the hospital WW isolates, *E. coli* 1000C-3 (Accession number CP091427), using long- and short-read sequencing. The genome contained 4,839,424 nucleotides with 50.75% GC content, 4,944 putative ORFs, and was identified as O101:H9 serotype, which has previously been associated with pathogenic and commensal strains [52]. It has also been found among calves with gastroenteritis [53]. Whole-genome sequencing identified this strain as sequence type (ST) 167, which has previously been associated with *bla_KPC_* encoding strains, and *Inc* types [54]. ST167 has been emerging globally, suggesting it as a high-risk clone [55]. In our study, 1000C-3 was resistant to SXT, GEN, CIP, TET, CTX, CRO, CAZ, meropenem (MEM), imipenem (IPM), and ertapenem (ERT) (Appendix A). The following resistance phenotypes were predicted using ResFinder based on the vast array of ARGs: aminocyclitol, macrolides, quaternary ammonium compounds, fluoroquinolones, phenicols, peroxides, aminoglycosides, tetracyclines, cephalosporins, and carbapenems. The chromosome encoded numerous ARGs involved in resistance to tetracycline (e.g., *tetB*), macrolides (e.g., *mdf(A)*), beta-lactams (e.g., *ampC*), and aminoglycosides (e.g., *kdpE*), as well an array of multidrug major facilitator superfamily (MFS) (Table 3: emrAB, emrKY) and resistance-nodulation-cell division (RND) efflux pumps (Table 3: AcrAB-TolC; AcrAD-TolC, MdtEF). These efflux pumps not only serve to export drugs and toxins but also aid in host colonization and dissemination through biofilm formation [56] and evasion of innate host defenses [57]. Additionally, *E. coli* 1000C-3 strain contained point mutations in *parC, parE,* and *gyrA*, providing quinolone resistance (Table 3). 

Beyond the resistance genotypes encoded chromosomally, *E. coli* 1000C-3 carried a mobilizable (pMHW-1: Accession number CP091429) and conjugative (pMHW-2) R plasmid. The smaller mobilizable plasmid, pMHW-1, was identified as an IncP-6 plasmid (Appendix A: 100% coverage and 99.88% identity to IncP-6 from *Serratia marcescens* accession number JF785550) with 40,232 base pairs and 57.87% GC content with 49 predicted ORFs, including carbapenemase *bla*_KPC-2_ (Figure 3 and Appendix A). The IncP-6 (assign to IncG in *E. coli*) replicon and *bla*_KPC-2_ have been identified amongst a variety of genera, including *Aeromonas* spp., *Klebsiella* spp., *Pseudomonas* spp., and *E. coli* acquired from environmental, animal waste, and clinical sources [54]. pMHW-1 shared over 99.8% identity with other plasmids sequences from *Pseudomonas aeruginosa*, *Citrobacter braakii* and *C. freundii, Klebsiella pneumoniae*, *Aeromonas veronii* and *A. hydrophila,* and *E. coli,* although only up to 89% coverage (Appendix A). The *bla*_KPC-2_ was just upstream of a truncated *bla*_TEM-1,_ located between IS*Kpn27* and IS*Kpn6*. This genetic arrangement: IS*Kpn6-bla*_KPC-2_-*bla*_TEM-1_-IS*Kpn27* has been evident in IncP6 from a variety of clinical and WW species [58]. pMHW-1 also carried transposon Tn5563, a member of the Tn3 transposon family, with a mercury operon encoding the periplasmic scavenger *merP,* transporter *merT,* and regulator *merR* (Figure 3). However, this strain lacks the detoxifying mercuric reductase *merA*, resulting in increased uptake of environmental mercury creating a hypersensitive strain [59]. Plasmid pMHW-1 appeared to be a mobilizable plasmid with an identifiable *oriT* site and relaxase gene (Figure 3 and Appendix A). It also encodes a replication initiation protein (RIP), as well as some genes aiding its replication and maintenance: *parA*, DNA polymerase III, *kfrA,* and *dnaC* (Appendix A). 

We also identified pMHW-2 (Accession number: CP091428), a multi-drug resistant self-transmissible IncFIA plasmid in *E. coli* 1000C-3 (Appendix A: 99.74% coverage and 99.48% identity with *E. coli* plasmid F Accession number AP001918) consisting of 145,445 bps with 52% GC content and 196 predicted ORFs. IncF plasmids are commonly associated with multi-drug resistant *E. coli* strains from the clinic, animals, and the environment [60]. This plasmid shared homology with numerous plasmids from *E. coli* (Appendix A). Conjugative pMHW-2 plasmid contained a *tra* operon with 30 genes aiding self-transmissibility (Figure 3 and Appendix A). These genes are involved in the regulation of transcription, relaxosome formation, type 4 secretion system, pilus formation, and mating pair formation [61]. pMHW-2 plasmid also contains a Tn5403, Tn3-like transposon, which has been identified to contain enzymes capable of helping non-conjugative plasmids [62]. 

The self-transmissible, multi-drug resistant plasmid pMHW-2 is rich in ARGs downstream of the type 1 integron integrase (Figure 3 and Appendix A: 14,767–36,470 bps). pMHW-2 encoded two beta-lactamases, *bla*_CTX-M-15_ and *bla*_OXA-1,_ with *bla*_CTX-M-15_ part of Tn2 between IS26 elements (Figure 3). Among clinical isolates, *bla*_CTX-M-15_ has been commonly identified on IncFI plasmids belonging to phylogroup A [63]. An IS*26*-composite transposon also contained *bla*_OXA-1_ along with type B-3 chloramphenicol O-acetyltransferase (*catB3*) and an aminoglycoside 6′-*N*-acetyltransferase-Ib (*aac(6’)-Ib-cr*) gene (Figure 3 and Appendix A). This plasmid also contained a macrolide resistance gene (*mph(A)*), tetracycline (*tetA*), sulphonamide (*sul1*), trimethoprim (*dfr17*), and aminoglycoside (*aadA5*), as well as resistance genes to disinfectants (e.g., *qacE* and *sitABCD*). The large presence of IS*26* interspersed within plasmids and chromosomes of clinical and environmental isolates within the *Enterobacteriaceae* family supports a pivotal role in their dissemination of multi-drug resistance [64,65]. Beyond resistance, this plasmid encodes an *iucABCD iutA* operon responsible for the biosynthesis of the iron siderophore aerobactin. The production of aerobactin has been associated with increased virulence and relapse episodes of urinary tract infections [66,67]. Under iron-limiting and oxidative stress conditions, aerobactin has increased biofilm formation and hydrogen peroxide resistance respectively [68]. When the whole-genome sequence was analyzed with PathogenFinder [69], *E. coli* 1000C-3 matched 663 pathogenic families, resulting in a pathogen probability of 0.919.

### 2.4. Prevalence of Virulence Genes among CTX-Resistant Populations 

Antimicrobial resistant opportunistic pathogens pose an even bigger health risk. Another way to assess the potential pathogenic nature of a bacterial isolate is to identify encoded virulence factors, which are involved in an array of activities including attachment to host tissues, immune evasion, and production of toxins capable of tissue damage [70]. Frequently, these virulence factors are found on pathogenicity islands and other mobile genetic elements that allow for gene transfer [71]. We analyzed seven virulence genes comprising both cell surface and secreted virulence factors [70]. CTX-resistant clinical strains exhibited significantly more virulence genes compared to WW effluent isolates (Figure 4A, *p* = 0.0007), where 64% of clinical isolates displayed three or more virulence genes compared to only 31% among WW effluent populations. Comparatively, the WW effluent isolates appeared to be less virulent; however, all encoded at least one virulence gene (Figure 4A). 

The virulence factor, aerobactin, was present in all clinical and effluent *E. coli* isolates. Aerobactin is frequently found among UPEC strains, is involved in iron sequestration, and is commonly associated with multi-drug resistance residing both on chromosomes and within plasmids [72]. Typically, aerobactin is associated with the presence of other virulence genes, such as type 1 fimbriae (*fimH*), hemolysin (*hlyCA*), and P fimbriae (*papC*) [73]. For our clinical strains, aerobactin and the adhesion gene, *fimH*, were co-expressed; similar findings were evident among effluent strains where co-expression occurred among 93% of the population (Figure 4B). Overall, virulence genes were significantly more abundant in the B2 phylogroup, regardless of source, compared to non-B2 phylogroups (Figure 4C, *p* < 0.05). This finding is consistent with other reports where phylogroup B2 has been associated with more virulence factors compared to other phylogroups [74,75]. In addition, we found that 75% of the B2 phylogroup isolates (from both clinical isolates and WW effluents) co-expressed aerobactin and afa adhesions (afa), a virulence factor commonly associated with UPEC [71]. However, due to the diversity and abundance of virulence genes in bacteria [76], it becomes difficult to attribute specific virulence genes to a pathogenic lineage, especially since they can be found among commensal strains [76]. Despite this limitation, our study detected similar virulence genes in both clinical samples and WW effluents including *afa* adhesions (Figure 4B) commonly associated with UPEC populations. These findings are consistent with a previous study identifying WW *E. coli* strains containing the same virulence factors as clinical UPEC isolates after whole-genome analysis [77]. Together, these findings implicate a potential pathogenic linkage emphasizing a health risk among treated effluent populations.

## 3. Materials and Methods

### 3.1. Acquisition of CTX-Resistant E. coli Isolates

WW samples were obtained from Jones Island Water Reclamation Facility (JI) and the Medical College of Wisconsin in Milwaukee, Wisconsin, USA. Facility employees acquired replicate 1L grab samples in sterile amber glass bottles (ThermoScientific, Waltham, MA, USA) once a week for three consecutive weeks in October 2018 from JI influent, treated effluent (post-primary and secondary treatment) and post-chlorinated disinfection effluents (final effluents entering environment). Hospital WW grab samples were acquired from two WW outflows at the Medical College of Wisconsin on two separate dates in October 2018. All samples were immediately placed on ice until it was processed within 8 h. Various water volumes (≤100 mL) were filtered over a 0.45 µm filter and placed on modified mTEC with cefotaxime (4 µg/mL) to obtain resistant *E. coli* populations. Random violet color colonies underwent further isolation by performing a four-quadrant streak on TSA with cefotaxime (2 µg/mL) to ensure isolation. From these plates, 58 WW isolates were obtained from hospital WW (*n* = 22), as well as influents (*n* = 19) and treated effluents (*n* = 17) with a maximum of 10 colony forming units (CFUs) per source per date. No CTX-resistant *E. coli* populations with the volumes tested were evident from one effluent sampling date and one hospital WW source on one date. Additionally, no CTX-resistant violet colonies were evident from post-chlorinated effluents in the volumes tested. Frozen glycerol stocks were made of all isolates for further analysis. 

Clinical *E. coli* isolates (e.g., urinary tract (7), blood (2), wound (1), and surgical tissue (1)) from inpatient (*n* = 5) and outpatient (*n* = 6) samples were acquired from the same medical facility as the hospital WW. Presumptive *E. coli* colonies were confirmed using Bruker MALDI-TOF mass spectrometry [19]. Potential ESBL- and carbapenemase-producing *E. coli* isolates were further identified using the BD Phoenix susceptibility system interfaced with the BD EpiCenter System V6.41 following Clinical & Laboratory Standards Institute (CLSI) guidelines. Frozen glycerol stocks were made of all isolates for future analysis. 

### 3.2. Antibiotic Susceptibility Testing 

The Kirby–Bauer disk diffusion method was used to determine the antibiotic resistance phenotypes of each isolate following CLSI guidelines [78]. Briefly, WW isolates were grown in tryptic soy broth overnight with cefotaxime (2 µg/mL). The susceptibility of all isolates was assessed after diluting overnight cultures to a MacFarland 0.5 standard followed by culturing them on Mueller Hinton plates stamped with the following antibiotics disks (BD BBL)—chloramphenicol (CHL, 30 mcg), sulfamethoxazole-trimethoprim (SXT, 23.75/1.25 mcg), ciprofloxacin (CIP, 5 mcg), gentamicin (GEN, 10 mcg), tetracycline (TET, 30mcg), ceftriaxone (CRO, 30 mcg), ceftazidime (CAZ, 30 mcg), ceftazidime/clavulanic acid (CAZ/CLAV, 30/10 mcg), cefotaxime (CTX, 30 mcg), meropenem (MEM, 10 mcg), imipenem (IMP, 10 mcg), and ertapenem (ERT, 10 mcg). Zones of inhibition were measured after 18–24 h incubation at 35 °C and susceptibility determined according to the 2020 CLSI standards M100-S25 [79]. ESBL producers were identified among CTX-resistant populations if the zone of inhibition with the CAZ/CLAV acid disk was ≥5 mm compared to the CAZ alone zone of inhibition.

### 3.3. Plasmid Isolation and Transformation 

Plasmids from CTX-resistant populations were isolated using a mini prep kit according to the manufacturer’s instructions (Zymo Research, Irvine, CA, USA). Isolated plasmid DNA was then separated on a 0.7% agarose gel at 80 V for 3 h. Gel images were captured after ethidium bromide staining for further analysis. 

To identify resistant phenotypes associated with plasmids, they were transformed into Lucigen E. cloni^®^ 10G chemically competent cells according to the manufacturer instructions (Lucigen, Middleton, WI, USA). Briefly, 4 µL of plasmid DNA was added to 40 µl of the competent cells and incubated on ice for 30 min. Heat shock was performed for 90 s at 42 °C followed by ice for 2 min. Recovery medium was added to the cells and incubated for 1 h at 37 °C while shaking at 175 rpm. The reaction was plated on cefotaxime (2 µg/mL) plates and incubated overnight at 35 °C. Transformants were further tested for antibiotic susceptibility to determine antibiotic resistant phenotypes transmissible via plasmids.

### 3.4. Identification of Beta-Lactamases and Virulence Genes

A Promega Genomic DNA Purification kit was used to isolate genomic DNA of resistant strains. β-lactamase genes were further amplified using 2X GoTaq Green PCR Master Mix (Promega Corp., Madison, WI, USA) and their associated primers (Appendix A). Isolated DNA from a Gram-negative carbapenemase detection panel (CDC & FDA AR Isolate Bank, Atlanta, GA, USA) served as positive controls for ESBLs. 

Detection of type 1 fimbriae (*fimH*), P fimbriae (*papC*), S and FIC fimbriae (*sfa*), Afa adhesin (*afa*), hemolysin (*hlyCA*), cytotoxic necrotizing factor (*cnf*), and aerobactin (*aer*) were all determined among clinical and treated effluent *E. coli* populations by PCR using primers listed in Appendix A according to previous methods [70]. 

### 3.5. Phylogroup Identification

Phylotyping was performed according to the revised Clermont method [25]. Briefly, a quadruplex PCR was performed using primers (Appendix A) specific to the following genes: *chuA*, *yjaA*, TspE4.C2, and *arpA* to determine its associated phylogroup. Further characterization of some phylogroup C and E strains was performed using two additional allele-specific PCR primer pairs (Appendix A) to *arpA* and *trpA* as described by Clermont et al. All PCR products were visible after gel electrophoresis using a 1.0% agarose gel with subsequent ethidium bromide staining. 

### 3.6. Whole Genome Sequencing and Analysis 

Genomic isolation was performed on *E. coli* 1000C-3 isolated from hospital WW that exhibited antibiotic resistance to third-generation cephalosporins and carbapenems. The Oxford Nanopore platform and NextSeq 2000 platform (Illumina, San Diego, CA, USA) were used to provide long and pair end reads (2 × 151 bp), respectively, for whole-genome sequencing. Illumina bcl2fastq 2.20.0.445 software and Porechop 0.2.3_seqan2.1.1 were performed on pair end reads and long reads, respectively, to perform quality control and adapter trimming. Unicycler 0.4.8 was subsequently used to create a hybrid assembly between the two reads and provide closed genomes. The WGS is available at NCBI under BioProject ID SUB10978992 with accession numbers CP091427-CP091429. 

The genome was submitted to JGI Integrated Microbial Genomes and Microbiomes (IMG/MER) for annotation and further analysis [80,81], including identifying putative horizontally transferred genes. Further analysis of assembled contigs were performed using the following web-based tools: PATRIC [82] for a comprehensive genome analysis; ResFinder 4.1 from the Center for Genomic Epidemiology [83] and Comprehensive Antibiotic Resistance Database (CARD) [84] to identify ARGs in the plasmid and chromosome; PlasmidFinder 2.1 and pMLST 2.0 to identify incompatibility types of plasmids [85]; MobileElementFinder v1.0.3 and IS Finder to identify mobile genetic elements [86,87]; SerotypeFinder 2.0 for serotyping [88]; oriTfinder 1.1 to identify oriT sites and MGEs [89]; and PathogenFinder 1.1 to determine the pathogenic potential [69]. Plasmid maps were comprised using the above tools and SnapGene software (from Insightful Science, San Diego, CA, USA; available at snapgene.com, accessed on 8 December 2021).

### 3.7. Statistics

Stata 12.1 software (College Station, TX, USA) and GraphPad Prism version 9.1.0 (GraphPad Software, San Diego, CA, USA) were used to perform all statistical tests. Fisher’s exact test was used to determine differences in the prevalence of ARGs and virulence genes among *E. coli* from various water sources. One-way ANOVA with Tukey’s post hoc test determined statistical differences between MAR indices. Correlations between antibiotic resistance phenotypes and genotypes were evaluated by Pearson’s correlation test. Variation among total virulence factors used Welch’s t-test.

## 4. Conclusions

With antibiotic resistance threatening global health, it is essential to understand the range and mechanisms of antibiotic resistance transmission [90] across diverse ecosystems. From this study, we can conclude the following: (1) hospital wastewater provides a rich source of multi-drug resistant *E. coli* populations and can serve as reservoirs of mobilizable ESBLs and carbapenemase entering sewer systems; (2) wastewater treatment and chlorine disinfection are critical mitigators in controlling AMR; (3) the prevalence and similarity of virulence factors among CTX-resistant *E. coli* in treated WW effluents are comparable to pathogenic strains in the same geographical area, further emphasizing the potential public health risk of treated wastewater into recreational waters. While this study demonstrates similarities in phenotypic and genotypic resistance across different ecosystems, additional sampling and more complete whole-genome sequencing would further support the evolutionary linkage between populations and enhance understanding the potential pathogenic nature of WW isolate populations.

## Figures and Tables

**Figure 1 antibiotics-11-00260-f001:**
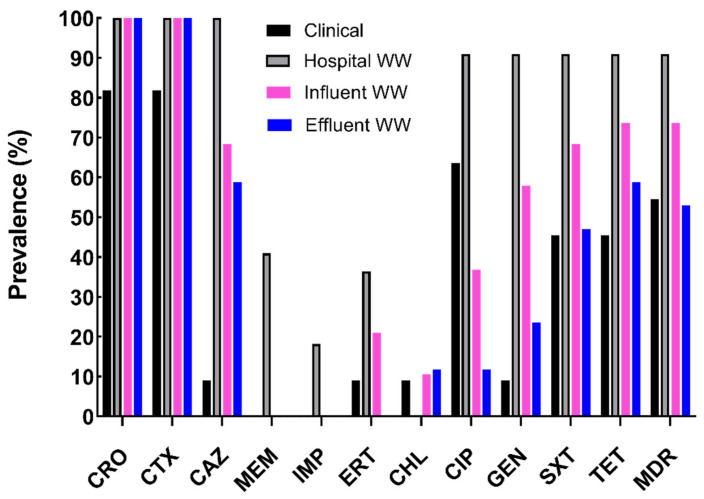
Prevalence of antibiotic resistance among *Escherichia coli* isolates. CTX-resistant clinical (*n* = 11, black) and wastewater (WW) isolates were analyzed for antibiotic susceptibility using disk diffusion. Isolates from wastewater populations included hospital (*n* = 22, gray), wastewater influent (*n* = 19, pink), and treated effluent (*n* = 17, blue). Prevalence was determined dividing resistant isolates by the total number of isolates in each source multiplied by a hundred. CRO: ceftriaxone; CTX: cefotaxime; CAZ: ceftazidime; MEM: meropenem; IMP: impenem; ERT: ertapenem; CHL: chloramphenicol; CIP: ciprofloxacin; GEN: gentamicin; SXT: sulfamethoxazole-trimethoprim; TET: tetracycline; MDR: multi-drug resistant.

**Figure 2 antibiotics-11-00260-f002:**
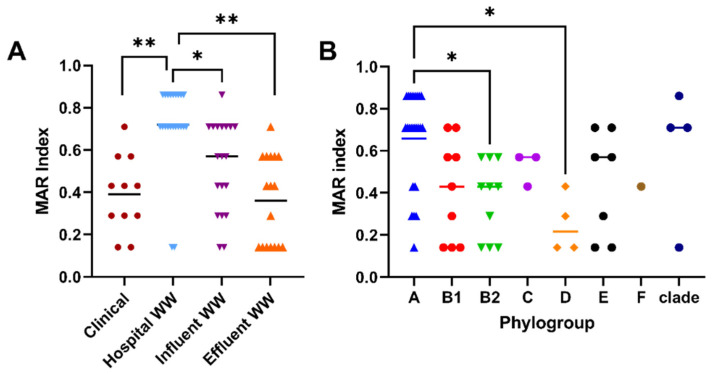
Antibiotic resistance among sample sources and phylogroups. (**A**) Multiple antibiotic resistance (MAR) index was determined by dividing the number of resistant antibiotic categories by the total number of antibiotic categories (*n* = 7) per isolate. (**B**) Phylogroup of total isolates was determined using the revised Clermont method. Each mark represents one isolate. The mean of each group is represented by a bar. Statistical differences between sources or phylogroups was determined using Tukey’s multiple comparison test. * *p* < 0.05; ** *p* < 0.001.

**Figure 3 antibiotics-11-00260-f003:**
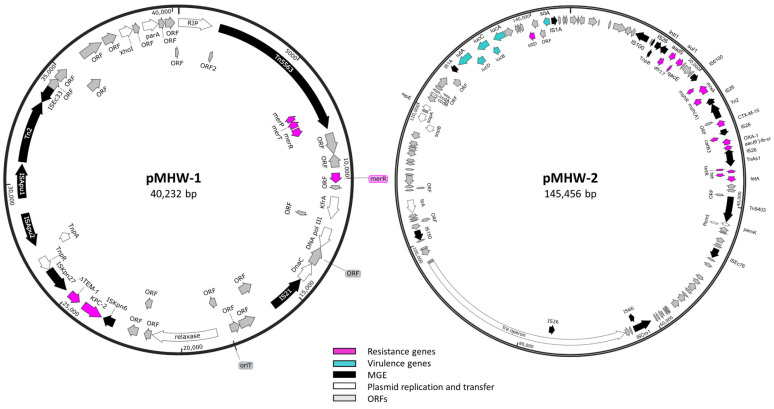
Plasmid maps from hospital wastewater *E. coli* 1000C-3 isolate. Plasmids sequences were determined from long-read Nanopore and Illumina short paired-end reads. pMHW-1 was identified as a 40,232 bps mobilizable IncP-6-type plasmid, and pMHW-2 as a 145,456 bps conjugative IncF plasmid containing a *tra* operon. Arrows are color-coded exhibiting genes involved in drug resistance (fuchsia), virulence (blue), mobile genetic elements (MGE) (black), plasmid replication, maintenance and transfer (white), and opening reading frames (ORFs) (grey).

**Figure 4 antibiotics-11-00260-f004:**
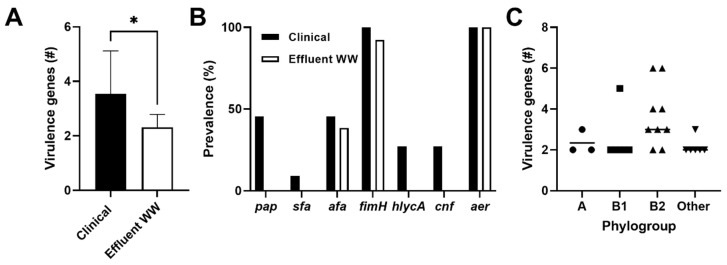
Virulence genes amongst CTX-resistant *E. coli* populations. (**A**) The mean number of virulence genes ± standard deviation identified among clinical *E. coli* (*n* = 11, black bars) and treated WW effluent populations (*n* = 16, white bars). (**B**) Data represents the prevalence of each virulence factor among clinical isolates (black bars) and WW treated effluents (white bars). Percent prevalence was calculated by dividing total isolates positive for each virulence factor by the total isolates from the representative source. (**C**) Each point represents the number of virulence genes encoded by an *E. coli* isolate within the major phylogroups with the bar representing the mean number of virulence genes within the corresponding phylogroup. * *p* = 0.0007.

**Table 1 antibiotics-11-00260-t001:** Prevalence of β-lactamases and *int1* genes among *E. coli* populations.

Source	N	Prevalence (%)							
		CTX-M-1 Subgroup	CTX-M-2 Subgroup	CTX-M-9 Subgroup ^a^	TEM	OXA	SHV	KPC ^b^	*int1*
Clinical	11	7 (64)	0	1 (9)	11 (100)	5 (45)	0	0	7 (64)
Hospital WW	22	21 (95)	0	0	13 (59)	19 (86)	0	9 (41)	17 (77)
Urban Influent	15	11 (73)	0	4 (27)	11 (73)	10 (67)	0	0	8 (53)
Treated effluent	16	13 (81)	0	5 (31)	12 (75)	13 (81)	0	0	9 (56)
Total	64	52 (81)	0	10 (16)	47 (73)	47 (73)	0	9 (14)	41 (64)

^a^ Statical difference between sources *p* = 0.005; ^b^
*p* < 0.001.

**Table 2 antibiotics-11-00260-t002:** Correlation of antibiotic resistance phenotypes and genotypes.

	CAZ	CRO	CTX	ERT	ESBL	*bla* _CTX-M-1_	*bla* _CTX-M-9_	*bla* _OXA_	*bla* _TEM_	*bla* _KPC_
CAZ	1									
CRO	0.1176 0.3548	1								
CTX	0.1176 0.3548	**1.0000 *** **0.0000**	1							
ERT	**0.2925 *** **0.0190**	−0.0949 0.4557	−0.0949 0.4557	1						
ESBL	**−0.3945 *** **0.0013**	0.2361 0.0604	0.2361 0.0604	**−0.3190 *** **0.0102**	1					
*bla* _CTX-M-1_	0.1732 0.1711	**0.4616** * **0.0001**	**0.4616** * **0.0001**	0.0066 0.9585	**0.2707** * **0.0305**	1				
*bla* _CTX-M-9_	**−0.4326** * **0.0004**	0.0954 0.4532	0.0954 0.4532	−0.1960 0.1205	0.0593 0.6417	−0.2343 0.0624	1			
*bla* _OXA_	0.1508 0.2342	0.0340 0.7897	0.0340 0.7897	−0.0076 0.9542	−0.1396 0.2713	−0.0170 0.8940	−0.1309 0.3024	1		
*bla* _TEM_	−0.2093 0.0969	0.0340 0.7897	0.0340 0.7897	**−0.2886** * **0.0207**	0.1440 0.2563	0.0736 0.5630	0.1614 0.2027	−0.2015 0.1103	1	
*bla* _KPC_	**0.3346** * **0.0069**	0.0897 0.4808	0.0897 0.4808	**0.6497** * **0.0000**	**−0.2505** * **0.0459**	0.1943 0.1239	−0.1741 0.1689	0.1415 0.2647	**−0.3673** * **0.0028**	1

Data from antibiotic susceptibility patterns against third-generation cephalosporins and carbapenems, as well as beta-lactamases, from all isolates were analyzed using Pearson’s correlation coefficients. The top value represents the correlation coefficient while the bottom number indicates the *p*-value. Values with statistically significant differences are in bold with an asterisk (*). ESBL: extended spectrum beta-lactamase.

**Table 3 antibiotics-11-00260-t003:** Resistome of MDR hospital wastewater *E. Coli* 1000C-3 isolate.

Antibiotic Resistance Genes	Location	Accession #	Resistance Target
*aac(6’)-Ib-cr*	Plasmid	DQ303918	Aminoglycoside and fluoroquinolone resistance
*aadA5*	Plasmid	AF137361	Aminoglycoside resistance
*aph(3’’)-Ib*	Plasmid	AF321551	Aminoglycoside resistance
*aph(6)-Id*	Plasmid	M28829	Aminoglycoside resistance
*blaCTX-M-15*	Plasmid	AY044436	Beta-lactam resistance
*blaOXA-1*	Plasmid	HQ170510	Beta-lactam resistance
*blaKPC-2*	Plasmid	AY034847	Beta-lactam resistance
*blaTEM-350*	Plasmid	WP045286946	Beta-lactam resistance
*mdf(A)*	Chromosomal	Y08743	Macrolide resistance
*mph(A)*	Plasmid	D16251	Macrolide resistance
*catB3*	Plasmid	AJ009818	Phenicol resistance
*sul1*	Plasmid	U12338	Sulphonamide resistance
*tet(A)*	Plasmid	AJ517790	Tetracycline resistance
*tet(B)*	Chromosomal	AF32677	Tetracycline resistance
*dfrA17*	Plasmid	FJ460238	Trimethoprim resistance
*sitABCD*	Plasmid	AY598030	Disinfectant resistance
*qacE*	Plasmid	X68232	Disinfectant resistance
*bacA*	Chromosomal	3002986	Peptide antibiotic resistance
*mdtG*	Chromosomal	3001329	MFS antibiotic efflux pump
*mdtH*	Chromosomal	3001216	MFS antibiotic efflux pump
*mdtP*	Chromosomal	3003550	MFS antibiotic efflux pump
*mdtN*	Chromosomal	3003548	MFS antibiotic efflux pump
*tolC*	Chromosomal	3000237	MFS antibiotic efflux pump
*emrA*	Chromosomal	3000027	MFS antibiotic efflux pump
*emrB*	Chromosomal	3000074	MFS antibiotic efflux pump
*emrK*	Chromosomal	3000206	MFS antibiotic efflux pump
*emrY*	Chromosomal	3000254	MFS antibiotic efflux pump
*evgA*	Chromosomal	3000833	MFS antibiotic efflux pump
*H-NS*	Chromosomal	3000676	MFS antibiotic efflux pump
*ampH*	Chromosomal	3004612	ampC-type beta-lactamase
*ampC1*	Chromosomal	3004611	ampC-type beta-lactamase
*cpxA*	Chromosomal	3000830	RND Efflux pump
*gadX*	Chromosomal	3000508	RND Efflux pump
*mdtE*	Chromosomal	3000795	RND Efflux pump
*mdtF*	Chromosomal	3000796	RND efflux pump
*acrA*	Chromosomal	3004043	RND Efflux pump
*acrB*	Chromosomal	3000216	RND Efflux pump
*acrD*	Chromosomal	3000491	RND Efflux pump
*baeS*	Chromosomal	3000829	RND Efflux pump
*baeR*	Chromosomal	3000828	RND Efflux pump
*marA*	Chromosomal	3000263	RND Efflux pump
*yojI*	Chromosomal	3003952	Peptide antibiotic efflux pump
*pmrF*	Chromosomal	3003578	Phosphoethanolamine transferase
*kdpE*	Chromosomal	3003841	Aminoglycoside efflux pump
*msbA*	Chromosomal	3003950	Nitroimidazole efflux pump
Point mutations		PMID	
parC AGC-> ATC		8851598	Quinolone resistance
parE TCG -> GCG		28598203	Quinolone resistance
gyrA TCG -> TTG		8891148	Quinolone resistance
gyrA GAC-> AAC		12654733	Quinolone resistance

The following ARGs and mutations were identified by ResFinder 4.1 from the Center for Genomic Epidemiology and Comprehensive Antibiotic Resistance Database. MFS: major facilitator superfamily; RND: resistance nodulation-cell division.

## Data Availability

The genomic sequence of *E. coli* 1000C-3 is available at NCBI under BioProject ID SUB10978992 with accession numbers CP091427-CP091429.

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
