# Peer review of "Similarities in Virulence and Extended Spectrum Beta-Lactamase Gene Profiles among Cefotaxime-Resistant Escherichia coli Wastewater and Clinical Isolates"

_antibiotics, 2022, doi:10.3390/antibiotics11020260_

Round 1

Reviewer 1 Report

Dear authors,

The research manuscript describes the investigation of the relationship between human and environmental cefotaxime resistant isolates with regards to the virulence factors and antibiotic resistance patterns. For a risk analysis on public health, the authors should include more samples and isolates and also information regarding the molecular sequence types of the isolates.

Missing data on the study settings:

  • The amount of of wastewater per day processed
  • From what settings the plant is collecting wastewater (agricultural activities, pharmaceuticals industries)
  • The inhabitants for the plant
  • The effluent is discharged into...
  • The river flows through the urban area?
  • The hospital wastewater is transported to the municipal WWTP without prior treatment?
  • From sampling, all colonies of E. coli were included in the analysis?
  • ESBL-producing E. coli were found in all the water samples.
  • The clinical isolates were collected from hospitalized or/and out-patients.

Importantly, the authors should highlight the importance of the study for the respective geographical area, there are any studies on human and environmental isolates performed before?

Thank you,

Author Response

Reviewer #1

Dear authors,

The research manuscript describes the investigation of the relationship between human and environmental cefotaxime resistant isolates with regards to the virulence factors and antibiotic resistance patterns. For a risk analysis on public health, the authors should include more samples and isolates and also information regarding the molecular sequence types of the isolates.

We would like to thank Reviewer #1 for their comments helping to make this a stronger manuscript. We addressed your comments in italics below: 

Further studies are ongoing to collect and analyze more isolates to better understand the risk analysis. However, considering these wastewater samples were all obtained in the same time period in the past, it would be difficult to address the reviewer’s comment in the work presented here. As a result, we revised the conclusion to bring awareness to this shortcoming and the need for whole genome sequencing as well.  

Missing data on the study settings:

  • The amount of wastewater per day processed
    • This has been included in lines 102 of revised manuscript
  • From what settings the plant is collecting wastewater (agricultural activities, pharmaceuticals industries)
    • This has been included on lines 102-103
  • The inhabitants for the plant
    • This has been included in lines 101
  • The effluent is discharged into...
    • The effluent is discharged into the Milwaukee Bay (43.02539471919148, -87.89543460250536) of Lake Michigan. This has been included on lines 103-105
  • The river flows through the urban area?
    • We are unsure what river you are alluding to here. There are three rivers that flow through the city and enter Lake Michigan at roughly the same site the effluent is released. The sewer pipes that arrive at Jones Island do collect from throughout the metropolitan area.
  • The hospital wastewater is transported to the municipal WWTP without prior treatment?
    • This is correct. No further treatment is performed once it enters the hospital sewer system
  • From sampling, all colonies of E. coli were included in the analysis?
    • We picked up to 10 cefotaxime-resistant isolates from each source on each date. However, some sampling dates from various sources did not have any cefotaxime-resistant isolates or less than ten colonies. These details have been updated on line 377-379  
  • ESBL-producing E. coli were found in all the water samples.
    • As was stated in the original manuscript line 88, we did not find any CTX-resistant isolates in the post-chlorinated wastewater. Also, as stated above, some of the sources did not have CTX-resistant isolates on every date. More details regarding this have been provided in the revised manuscript lines 377-380.
  • The clinical isolates were collected from hospitalized or/and out-patients.
    • These samples were from both inpatient (n=5) and outpatients (n=6). These details have been provided on line 383

Importantly, the authors should highlight the importance of the study for the respective geographical area, there are any studies on human and environmental isolates performed before?

We agree and have included in the introduction. Our labs have characterized resistant phenotypes and genotypes in the past. The novelty in this manuscript is the focus cefotaxime-resistant populations throughout this similar geographical regions and timeframe.  

Thank you,

Reviewer 2 Report

The manuscript by Liedhegner et al. describes E. coli strains isolated from 4 different sources. The authors characterise the strains focusing in particular on their AMR profiles and they compare resistance patterns of the isolates from different sources. Resistance of  Enterobacteriaceae to 3rd and 4th generation cephalosporins and to carbapenems is an important topic and has been a focus of a great number of publications in the recent years. This manuscript describes a small scale study from a geographically restricted area. The manuscript is well written, the data are clearly presented and discussed in sufficient detail.

The main weakness of the study is the use of PCR based method to identify the antimicrobial resistance genes, while these days majority of studies use WGS for this purpose. The primer set e.g. contains primers for only few CTX-M genes and therefore may not identify or discriminate between other gene groups not covered by the primer set. NB. the supplementary file does not contain the citations for the referenced primer sets and therefore it was impossible to verify the range of genes amplified by the listed primers so please add the references section to the supplementary file.

Major comments:

Throughout the manuscript the authors  refer to some beta-lactamase genes as ESBL, which is not always correct. E.g. TEM-1 enzymes are not ESBL, but broad spectrum beta-lactamases. Similarly, not all blaOXA genes code for ESBL. To date hundreds of different variants of beta-lactamase genes have been described, and a large proportion of these genes are not ESBL. That is likely the reason why the authors do not find a correlation between ESBL phenotype and the presence of blaTEM genes. Considering that the authors did not identify the exact variants of some of the bla genes it is crucial that they correct the manuscript and tables accordingly.

Other comments:

Please add information about usage of 3rd generation cephalosporins in United States.

The last paragraph of Introduction, please re-write this part to clarify why this study was performed. The authors list three aims but they do not mention the significance of the study. 

L76 - please correct the part 'antibiotic threats facing healthcare' - I believe authors meant antimicrobial resistance threats?

Materials and methods: please add the version numbers of all the bioinformatic tools used and the versions (or date and number of genes in the database) for all databases used. Make sure all tools are properly referenced (e.g. Unicycler)

Conclusions: Considering that the authors combined Result and Discussion sections, the Conclusion section should summarise all the major findings from the study and their possible implications. The authors should also mention any possible weaknesses of their work that they identified. Moreover, the authors state that this study show how different ecosystems are connected (via AMR patterns), however the actual connections cannot be inferred from this work. Except of identifying the phylogroups, the authors did not conduct any type of phylogenetic study and therefore they cannot speculate whether or not the strains isolated from different sources are genetically related.  The authors should therefore concentrate their conclusion on the actual findings.

Author Response

Reviewer #2

The manuscript by Liedhegner et al. describes E. coli strains isolated from 4 different sources. The authors characterise the strains focusing in particular on their AMR profiles and they compare resistance patterns of the isolates from different sources. Resistance of  Enterobacteriaceae to 3rd and 4th generation cephalosporins and to carbapenems is an important topic and has been a focus of a great number of publications in the recent years. This manuscript describes a small scale study from a geographically restricted area. The manuscript is well written, the data are clearly presented and discussed in sufficient detail.

We would like to thank Reviewer #2 for recognizing the importance of the subject matter and providing their valuable comments making this a stronger manuscript. We have addressed the Reviewer’s comments below in italics:

The main weakness of the study is the use of PCR based method to identify the antimicrobial resistance genes, while these days majority of studies use WGS for this purpose. The primer set e.g. contains primers for only few CTX-M genes and therefore may not identify or discriminate between other gene groups not covered by the primer set. NB. the supplementary file does not contain the citations for the referenced primer sets and therefore it was impossible to verify the range of genes amplified by the listed primers so please add the references section to the supplementary file.

We apologize for the absence of these citations and they have been included in the revision. We agree with the use of WGS; however, this would require increased funding for the study which was not available.

Major comments:

Throughout the manuscript the authors refer to some beta-lactamase genes as ESBL, which is not always correct. E.g. TEM-1 enzymes are not ESBL, but broad spectrum beta-lactamases. Similarly, not all blaOXA genes code for ESBL. To date hundreds of different variants of beta-lactamase genes have been described, and a large proportion of these genes are not ESBL. That is likely the reason why the authors do not find a correlation between ESBL phenotype and the presence of blaTEM genes. Considering that the authors did not identify the exact variants of some of the bla genes it is crucial that they correct the manuscript and tables accordingly.

We agree with the Reviewer and appreciate bringing our awareness to this issue. We have scanned the whole manuscript and changed the applicable texts to read b-lactamases when talking about blaOXA and blaTEM

Other comments:

Please add information about usage of 3rd generation cephalosporins in United States.

This has been added on pg 54-55

The last paragraph of Introduction, please re-write this part to clarify why this study was performed. The authors list three aims but they do not mention the significance of the study. 

Has been included

L76 - please correct the part 'antibiotic threats facing healthcare' - I believe authors meant antimicrobial resistance threats?

Thank you for catching this. It has been corrected.

Materials and methods: please add the version numbers of all the bioinformatic tools used and the versions (or date and number of genes in the database) for all databases used. Make sure all tools are properly referenced (e.g. Unicycler)

We have updated and posted the version numbers of all the bioinformatic tools

Conclusions: Considering that the authors combined Result and Discussion sections, the Conclusion section should summarise all the major findings from the study and their possible implications. The authors should also mention any possible weaknesses of their work that they identified. Moreover, the authors state that this study show how different ecosystems are connected (via AMR patterns), however the actual connections cannot be inferred from this work. Except of identifying the phylogroups, the authors did not conduct any type of phylogenetic study and therefore they cannot speculate whether or not the strains isolated from different sources are genetically related.  The authors should therefore concentrate their conclusion on the actual findings.

The conclusion has been rewritten to address the Reviewer’s comments

Reviewer 3 Report

The manuscript by Liedhegner et al describes “Similarities in Virulence and Extended-Spectrum Beta-Lactamase Gene Profiles among Cefotaxime-Resistant Escherichia coli Wastewater and Clinical Isolates”.In this study, authors compared antibiotic resistance patterns between ESBL-producing Escherichia coli from human clinical diseases and cefotaxime-resistant environmental strains, as well as their potential to be pathogenic. Authors compare antibiotic susceptibility between clinical isolates, hospital wastewater, and urban wastewater and they observed Multi-drug resistance predominated among hospital and urban wastewater influent isolates. They also performed whole-genome sequencing of a carbapenemase-producing hospital wastewater E. coli strain which revealed plasmid-mediated blaKPC-2.

Overall, the manuscript is well organized, and the information was explored to show what they intend to show. To my knowledge, the integrated application of these biochemical and genome sequencing approaches is something very interesting and could deliver novel data for the future identification of virulent strains. The introduction is well written, and the material and methods are described adequately. Results and discussion part are properly described, and the technological approaches are rationally developed. However, a few minor weaknesses are noted that can be easily addressed before the manuscript is accepted for publication.

Comments: It would be better to explain ESBL in the abstract because you use first time in this section. Some references did not follow the journal pattern like 17,18,19,27 etc.

Author Response

Reviewer #3

The manuscript by Liedhegner et al describes “Similarities in Virulence and Extended-Spectrum Beta-Lactamase Gene Profiles among Cefotaxime-Resistant Escherichia coli Wastewater and Clinical Isolates”.In this study, authors compared antibiotic resistance patterns between ESBL-producing Escherichia coli from human clinical diseases and cefotaxime-resistant environmental strains, as well as their potential to be pathogenic. Authors compare antibiotic susceptibility between clinical isolates, hospital wastewater, and urban wastewater and they observed Multi-drug resistance predominated among hospital and urban wastewater influent isolates. They also performed whole-genome sequencing of a carbapenemase-producing hospital wastewater E. coli strain which revealed plasmid-mediated blaKPC-2.

Overall, the manuscript is well organized, and the information was explored to show what they intend to show. To my knowledge, the integrated application of these biochemical and genome sequencing approaches is something very interesting and could deliver novel data for the future identification of virulent strains. The introduction is well written, and the material and methods are described adequately. Results and discussion part are properly described, and the technological approaches are rationally developed. However, a few minor weaknesses are noted that can be easily addressed before the manuscript is accepted for publication.

Comments: It would be better to explain ESBL in the abstract because you use first time in this section. Some references did not follow the journal pattern like 17,18,19,27 etc.

We thank you for your critical review and recognizing the significance of the manuscript. Regarding the ESBL definition, the abstract is limited to only 200 words, so we moved the definition of ESBL from the results/discussion section to the introduction (line 59-61).

We have also scanned through the references and removed any “https” uses, ensured all bacteria were italicized, and reformatted using Endnote. Unfortunately, we are not exactly sure what part of the reference was inaccurate. Any extra details would be greatly appreciated and apologize for any unintentional formatting issues.  

Round 2

Reviewer 1 Report

I thank the authors for their very detailed replies to comments on the submitted version.